# Misconceptions and Behavioral Risks in Parental Antibiotic Use on Romanian Children: A Cross-Sectional Study on Knowledge, Attitudes, and Practices

**DOI:** 10.3390/antibiotics14050479

**Published:** 2025-05-09

**Authors:** Alin Iuhas, Radu Galiș, Marius Rus, Andreea Balmoș, Cristian Marinău, Larisa Niulaș, Zsolt Futaki, Dorina Matioc, Cristian Sava

**Affiliations:** 1Faculty of Medicine and Pharmacy, University of Oradea, 410087 Oradea, Romania; 2Bihor County Clinical Emergency Hospital, 410167 Oradea, Romania

**Keywords:** antibiotic resistance, antibiotic consumption, antimicrobial stewardship, children, caregiver perception, Romania

## Abstract

Background: Antimicrobial resistance is a growing global health threat, with antibiotic misuse in pediatric populations being a significant contributing factor. In Romania, antibiotic consumption and resistance rates are among the highest in Europe. Objective: To assess Romanian parents’ knowledge, attitudes, and practices regarding antibiotic use in children, and to identify key misconceptions and behavioral risks contributing to inappropriate antibiotic use. Methods: A cross-sectional survey was conducted among 400 parents of hospitalized children in a pediatric department in Romania. Participants completed a 15 item structured questionnaire. Data were analyzed using descriptive statistics, chi-square tests, and binary logistic regression to examine associations and control for potential confounding effects between education level, residential environment, and parental misconceptions regarding antibiotic use. Results: Among the 400 surveyed caregivers, 86% (*n* = 344) held at least one misconception regarding antibiotic use. Additionally, 42.5% (*n* = 170) of participants reported that they had never heard of the concept of antibiotic resistance. Misconceptions were significantly more prevalent among individuals with lower levels of education and those residing in rural areas (*p* < 0.001). While 89.8% (*n* = 359) stated that they had never administered antibiotics to their children without a physician’s recommendation, a separate subset of 28% (*n* = 112) acknowledged that they had asked a doctor to prescribe antibiotics for their child. Moreover, 23.3% (*n* = 93) reported seeking a second medical opinion when antibiotics were not initially prescribed. Conclusions: Despite high adherence to medical advice, widespread misconceptions persist. These findings highlight the need for targeted, population-specific educational interventions to promote rational antibiotic use and address AMR in high-burden settings like Romania.

## 1. Introduction

Antimicrobial resistance (AMR) is an urgent global public health problem with significant global economic and security implications [1,2,3]. The consequences of antibiotic resistance are far-reaching, extending beyond individual patients to the entire healthcare system, leading to prolonged hospital stays, increased medical costs, and higher mortality rates [4], with an estimated 4.95 million worldwide deaths associated with bacterial AMR in 2019 [5]. AMR is projected to have a devastating impact over the coming decades. By 2030, it could push 24 million people into extreme poverty and by 2050, AMR is estimated to cause 10 million deaths annually and result in economic losses reaching $100 trillion, affecting healthcare costs, labor markets, global trade, and poverty [6].

Effectively addressing antimicrobial resistance requires a multifaceted strategy encompassing stringent infection prevention and control (IPC) measures to mitigate the transmission of resistant bacteria, comprehensive antimicrobial stewardship programs (ASPs), and robust surveillance systems [1,2,3]. Although evidence suggests that well-implemented ASPs and IPC programs in hospitals can effectively reduce the burden of antibiotic resistance, the rising prevalence of preventable healthcare-associated infections indicates that many healthcare institutions either underestimate the issue or struggle to address this challenge adequately [7].

Antimicrobial consumption (AMC) is one of the primary factors contributing to the development of AMR [1]. AMC patterns across European Union/European Economic Area (EU/EEA) countries have exhibited significant differences in how well they align with the World Health Organization (WHO) targets [1,2]. A distinct AMC and AMR gradient is observed, exhibiting a clear north-to-south and west-to-east pattern. Lower resistance rates are predominantly reported in northern and western regions, whereas higher resistance levels are more frequently observed in the eastern and southern regions [2]. AMR is positively correlated with AMC [8]. Romania is currently facing a public health challenge due to its simultaneously high levels of AMC and AMR. According to recent surveillance data, the country ranks among the top in the European Union for both community and hospital antibiotic use and exhibits some of the highest rates of resistance to commonly prescribed antibiotics. This dual burden not only complicates infection management locally but also poses a risk for the wider European region through potential cross-border transmission and reduced effectiveness of antimicrobial therapies [9,10,11], with little information available about antibiotic resistance in community-acquired infections or in veterinary settings [9]. Furthermore, the Romanian healthcare system has the highest antibiotic consumption (community and hospital sectors combined) in Europe, a situation exacerbated by limited awareness among both healthcare providers and the general public, with a growing trend over recent years [12].

According to the Special Eurobarometer 478 on Antibiotic Resistance, public understanding of proper antibiotic use in Romania remains alarmingly low. Only 37% of Romanians correctly recognize that antibiotics are ineffective against viral infections, highlighting a significant knowledge gap. Furthermore, 83% of respondents reported not receiving any information in the past 12 months regarding the unnecessary use of antibiotics, indicating a deficiency in public health communication. Alarmingly, one in four individuals believes that antibiotic treatment can be discontinued once symptoms improve, rather than completing the full course as prescribed. In addition, the Eurobarometer reports that only 40% of Romanian respondents stated they had received antibiotics based on a diagnostic test, suggesting that clinical decision-making practices may not consistently rely on microbiological confirmation [13].

Children are especially vulnerable to infections and are often prescribed antibiotics [14,15]. A critical issue in pediatric antimicrobial use is the misalignment between prescription practices and clinical guidelines. Studies have shown that a substantial proportion of pediatric antibiotic prescriptions in both hospital and outpatient settings are inappropriate, either in terms of indication, dosage, or duration [16,17,18]. This problem is particularly prevalent in primary care, where broad-spectrum antibiotics are often prescribed unnecessarily due to diagnostic uncertainty or parental pressure [19]. Furthermore, in many low- and middle-income countries, limited access to rapid diagnostic tools leads to empirical antibiotic use, further exacerbating AMR risks [20,21].

Parents, as the primary decision-makers in the procurement and administration of medications, play a crucial role in ensuring the appropriate use of antibiotics in pediatric care [22]. Often driven by misconceptions and a lack of knowledge, caregivers’ misuse of antibiotics significantly contributes to the problem. Studies show that despite the public campaigns, patients believe antibiotics can treat viral infections like the common cold or flu, or act as substitutes for anti-inflammatory drugs [23,24]. The accessibility and availability of antibiotics without prescription in some regions further exacerbate the issue [25]. Caregivers, especially in in remote areas, with limited access to health services, may rely on advice from their networks when deciding to use antibiotics for perceived health threats to their children [22].

The overuse and misuse of antibiotics in children not only contribute to AMR but also increase the risk of adverse drug reactions, disruption of the gut microbiome, and long-term health implications [26,27].

Understanding parental perceptions and behaviors regarding antibiotic use is essential to addressing this issue effectively. The aim of this study is to explore the knowledge, attitudes, and practices of Romanian caregivers concerning antibiotic use in children, with a focus on identifying key misconceptions and barriers to proper antibiotic adherence. The findings will provide valuable insights into the extent of inappropriate antibiotic use and serve as a basis for developing targeted interventions to improve awareness and promote responsible antibiotic use.

## 2. Results

The study included a total of 400 participants who fully completed the questionnaire. Participants who declined to provide demographic information were included, provided that the questionnaire section was fully completed. The vast majority of respondents were female (*n* = 364, 91%), while the remaining 9% (*n* = 36) were male. The mean age of the respondents was 33.98 years, with a median of 34 years. The youngest participant was 14 years old, while the oldest was 68 years old. A small proportion of participants (*n* = 15, 3.8%) chose not to disclose their age.

The majority of participants had attained either secondary education (high school or upper-level schooling, 40.8%) or tertiary education (higher education or university-level studies, 32.8%), while 25.8% reported only primary or no formal education (elementary or basic education). Additionally, 0.8% of respondents chose not to disclose their educational background.

Participants were nearly evenly distributed between rural (47.3%) and urban areas (47.3%), while a smaller proportion (4.8%) resided in suburban areas—defined as rural localities adjacent to urban centers with similar levels of access to information. Additionally, 0.8% of respondents chose not to disclose their place of residence.

The questionnaire consisted of 15 close-ended questions, allowing participants to respond with “Yes”, “No”, or “I don’t know”. The analysis provides insight into caregivers’ awareness, beliefs, and practices concerning antibiotic use in children. In response to the first question, “Have you ever heard of the term antibiotic resistance?”, 230 respondents (57.5%) answered affirmatively, while the remaining 42.5% indicated they were unfamiliar with the concept. When asked whether antibiotics should be routinely administered for viral infections, only 55% (*n* = 220) responded correctly with “No”, whereas 21% (*n* = 84) believed antibiotics should be used, and 24% (*n* = 96) were uncertain. A notable minority proportion of caregivers (35.8%, *n* = 143) admitted to having administered antibiotics to their child during a common cold in an attempt to prevent symptom progression; meanwhile, 5.8% (*n* = 23) were unsure, and 58.5% (*n* = 234) reported they had not engaged in this practice. Regarding the belief that symptoms such as sore throat or productive cough always require antibiotics, 25.5% of participants responded affirmatively, while 61.8% correctly denied this assumption. A similar trend was observed when asked whether the presence of fever necessitates antibiotics, with 63.5% (*n* = 235) rejecting this notion, 24.3% (*n* = 97) affirming it, and 12.3% remaining uncertain. The majority of respondents (89.8%, *n* = 359) stated that they had never administered antibiotics without a physician’s recommendation. However, 28% (*n* = 112) admitted to having asked a physician to prescribe antibiotics for their child, and 23.3% (*n* = 93) reported seeking a second opinion when a doctor declined to prescribe antibiotics. Despite this, most participants (71.3%, *n* = 285) reported accepting the physician’s decision, and 87.2% (*n* = 312) stated that they generally follow the doctor’s advice when antibiotics are prescribed. Only 39.1% (*n* = 140) of respondents correctly identified that not all tablets can be crushed for easier administration. In contrast, 15.6% (*n* = 56) incorrectly believed this practice is universally acceptable, and 45.3% (*n* = 162) reported not knowing. When asked whether they reuse leftover antibiotic suspensions, 68.2% (*n* = 244) responded “No”, while the remaining respondents either admitted to doing so or were unsure. Regarding treatment adherence, 83.8% (*n* = 300) stated that they had never discontinued an antibiotic course prematurely, even when their child began to feel better. Furthermore, 68.4% (*n* = 245) reported that they had never refused a prescribed antibiotic on the grounds that it was not suitable or because they believed only certain antibiotics work for their child. When asked about reading medication information leaflets, 84.9% (*n* = 304) reported that they routinely read them. Finally, 80.2% (*n* = 287) of participants stated that they had never obtained antibiotics from a pharmacy without a prescription, in line with current national regulations (Figure 1).

Both bivariate and multivariate analyses indicate that educational level plays a critical role in the prevalence of misconceptions related to antibiotic use. A chi-square test of independence revealed statistically significant associations between both education level and environment of origin and the presence of at least one misconception. Participants with tertiary education were significantly less likely to hold misconceptions (65.6%) compared to those with secondary (94.5%) or primary/no education (98.1%), χ^2^(3) = 67.72, *p* < 0.001. Similarly, respondents from urban areas were less prone to misconceptions (79.4%) than those from rural (92.1%) or sub-urban environments (89.5%), χ^2^(3) = 13.36, *p* = 0.004. These findings were further supported by a binary logistic regression model, which confirmed that education level was a strong independent predictor of misconceptions (Nagelkerke R^2^ = 0.279; *p* < 0.001), while environment of origin did not remain significant when controlling for education. Specifically, participants with only primary or no education were nearly 23 times more likely (OR = 22.93, *p* < 0.001), and those with secondary education over eight times more likely (OR = 8.26, *p* < 0.001) to exhibit at least one misconception compared to those with tertiary education. In contrast, environment of origin did not significantly predict misconception presence in the adjusted model (*p* = 0.759).

## 3. Discussion

Although the dangers associated with antimicrobial resistance have been evident and widely discussed in the scientific literature for decades, surveys and national reports have indicated that awareness of AMR is relatively low among both the general public and, at times, healthcare professionals [24,25]. Pediatric patients, one of the most vulnerable groups, rely entirely on their caregivers’ understanding of antibiotic use for their present and future health.

This study provides novel data from Romania, a country identified as having one of the highest levels of antimicrobial resistance (AMR) in Europe—a region that remains underrepresented in current international research. The findings contribute valuable insight into the sociocultural and behavioral dimensions of antibiotic use in a context where both public health challenges and access to accurate health information remain uneven. Romania’s unique combination of high AMR prevalence, significant urban–rural disparities, and varied educational attainment among caregivers makes it a critical setting for examining how knowledge, misconceptions, and healthcare behaviors intersect. By focusing on the beliefs and practices of parents and caregivers of hospitalized children, this study adds to the growing body of evidence emphasizing the need for targeted, context-specific interventions in AMR prevention and antibiotic stewardship.

This study originates from empirical observations of misconceptions among parents regarding antibiotic therapy and aims to objectively assess these misunderstandings. To achieve this, we designed a 15 question survey, structured into three main categories: 1—Knowledge and Misconceptions—evaluating parental misunderstandings about antibiotics, as observed empirically in the patient population served by our hospital.; 2—Misuse/Self-Medication—assessing the extent of self-prescribed antibiotic use, a practice that, despite governmental efforts to limit it, still appears to be prevalent in our region; 3—Compliance with Treatment—investigating adherence to prescribed antibiotic regimens, indirectly reflecting the trust that patients and caregivers place in their physicians and the healthcare system. These three categories will be discussed separately.

### 3.1. Knowledge and Misconceptions About Antibiotics

The first set of questions (Q1, Q2, Q3, Q4, Q5, Q14, and Q15) assessed parental understanding of antibiotic use and common misconceptions. A key finding was the proportion of parents (42.5%) who were unaware of the concept of antibiotic resistance (Q1). This lack of awareness suggests that public health campaigns may not have effectively reached or educated parents about the long-term consequences of antibiotic misuse.

A significant number of respondents (45%) are not aware that antibiotics should not be routinely administered for viral infections (Q2) or that symptoms such as sore throat, cough with mucus, and fever (Q4, Q5) do not always require antibiotic treatment. These misconceptions reflect a fundamental misunderstanding of the difference between bacterial and viral infections, which can lead to unnecessary antibiotic use and contribute to AMR.

A significant number of respondents (*n* = 143, 35.8%) stated that they had administered antibiotics for a viral infection in an attempt to prevent symptom worsening (Q3). This behavior highlights a misplaced belief in the prophylactic role of antibiotics, which may be fueled by misinformation, prior medical experiences, or a lack of clear guidance from healthcare professionals.

Furthermore, some parents were unaware of proper medication handling practices, as indicated by their responses to Q14 and Q15. The belief that any type of tablet can be crushed for easier administration is present in 15.6%, with another 45.3% that do not know if the tablets can be safely crushed or not; and that leftover antibiotic suspensions can be stored for future use was present in 18.4%, with another 13.4% that are not sure if this is a safe practice. These findings emphasize the need for targeted educational interventions to address these critical gaps in knowledge.

Overall, 86% of respondents (*n* = 344) exhibited some form of misconception regarding antibiotic use. These misconceptions are widespread and can be found to varying degrees across different populations [26,27,28,29,30,31,32]. Even in populations where the majority of the population has accurate knowledge of the indications for antibiotics, decisions regarding the necessity of antibiotics are often influenced by fear of severe illness [22]. Knowledge that antibiotics are not effective against viral infections is limited in Romania, at 37.0%, compared with a 43.0% average across 29 EU member states. Greece at 23.0%, has lowest level among EU states and Sweden at 74.0% has the highest proportion of population with accurate antibiotic knowledge [13].

Multiple studies have correlated a higher level of education, through better access to information, with a higher level of knowledge and awareness regarding AMR. Poorer areas, with individuals having lower levels of education, are more prone to misinformation regarding the need for antibiotic treatment [33,34,35]. This highlights the need for targeted informational campaigns aimed at these populations [36]. Recent studies have explored the relationship between education level, environment of origin, and misconceptions about antibiotic use and antimicrobial resistance. A systematic review and meta-analysis in 2022 examined the association between education level and antibiotic misuse. The study found that individuals with higher education levels (>12 years) had 14% lower odds of misusing antibiotics compared to those with lower education levels (≤9 years). However, the confidence intervals were broad, suggesting variability in the data. Interestingly, higher education was associated with a 41% increase in the likelihood of storing antibiotics for future use, indicating that while education may reduce certain forms of misuse, it might contribute to others. The study also noted that in high-income countries, individuals with medium education levels had 20% lower odds of antibiotic misuse compared to those with low education levels [37]. In Bosnia and Herzegovina, a cross-sectional study by Glibić et al. in 2023 compared urban and rural populations regarding their knowledge and behavior about antibiotic use. Urban participants demonstrated better knowledge and had higher education levels than their rural counterparts. Women in urban areas, in particular, exhibited significantly better understanding. Despite adequate knowledge among many respondents, improper antibiotic use was prevalent, especially in rural areas where self-medication was more common. The study concluded that having a medical professional in the family was linked to better knowledge about antibiotics, whereas the general educational level was not a significant factor. These studies highlight the complex interplay between education, environment, and misconceptions about antibiotic use. They underscore the need for tailored educational programs that address specific misconceptions prevalent in different demographic groups [38].

In Romania, the general perception among patients is that they do not receive sufficient information regarding the unnecessary use of antibiotics, as 83% of respondents reported not recalling any such information in the past 12 months [13]. Among those who did receive information, the majority (69%) obtained it from healthcare professionals or medical facilities [13].

Interestingly, a study from Japan found that 80% of participants were unaware that antibiotics do not eliminate viruses. However, Japan has reported a decline in antibiotic consumption [39,40], suggesting that factors beyond knowledge, such as stricter regulations, improved prescribing habits, or cultural influences, contribute to more rational antibiotic use.

### 3.2. Self-Medication and Unregulated Antibiotic Use

The second group of questions (Q3, Q6, Q12, and Q13) focused on self-medication practices among parents. The frequency with which parents administered antibiotics to their children without a doctor’s recommendation (Q6) was found to be low (8.8%). Self-medication with antibiotics is a major driver of AMR, as incorrect dosages or inappropriate antibiotic choices can promote bacterial resistance.

Another concerning aspect was the preventive use of antibiotics for common colds (Q3). Although this practice was discussed in the subsection addressing misconceptions and misinformation, the use of leftover antibiotics for presumed prophylaxis in viral infections remains widespread in our population, as also showed in a 2018 study exploring the primary drivers of irrational antibiotic use in Europe [41]. In some cases, this behavior is further reinforced by healthcare professionals, either through past prescribing habits or the common practice of prescribing antibiotics to be used only if symptoms worsen [21]. However, in the absence of a follow-up consultation to assess the necessity of antibiotic initiation, this approach may contribute to unjustified antibiotic use and increased antimicrobial resistance. The fear of severe illness, particularly among patients with chronic conditions that could be exacerbated by an infection, often drives caregivers and even healthcare providers to add antibiotics to the treatment regimen—sometimes unjustifiably [21,42,43,44].

The accessibility and availability of antibiotics without prescription in some regions further exacerbate the issue [41]. The growing issue of using antibiotics without a prescription, whether purchased online or in another country, is highlighted in a European survey [44]. This practice of self-medication increases the risk of improper dosing, incomplete treatment courses, and ultimately, contributes to the growing problem of resistance. Furthermore, the premature discontinuation of antibiotic treatment once a child’s symptoms improve is another common behavior that fuels antibiotic resistance, allowing surviving bacteria to develop resistance mechanisms and propagate. In our study, 18% of parents reported that they were able to purchase antibiotics without a prescription (Q12). This suggests that pharmacies may still be dispensing antibiotics illegally, despite existing regulations. Strengthening enforcement mechanisms to prevent over-the-counter antibiotic sales is crucial in tackling self-medication. Romanian legislation mandates that antibiotics can only be dispensed with a medical prescription, with the exception of emergency situations where immediate access to a doctor is not possible. Additionally, the sale of antibiotics online is strictly forbidden. At the beginning of 2024, this regulation was further strengthened by Ministerial Order No. 63/2024, which introduced a national framework for monitoring the prescription and dispensing of antibiotics and antifungals. The new provisions allow pharmacists to dispense a limited emergency dose (covering up to 48 h of treatment) without a prescription, provided that the patient or caregiver signs a declaration of responsibility, commits to seeking medical consultation as soon as possible, and provides details about the symptoms justifying the request. These measures aim to balance accessibility to essential medication in urgent cases while maintaining strict control over antibiotic use to combat antimicrobial resistance [45,46]. Our local findings show a slightly higher percentage of patients obtaining antibiotics without a prescription compared to the national estimates, where 84% of respondents in the survey stated that they obtained their last course of antibiotics from a medical practitioner [13].

Interestingly, responses to Q13 (Do you usually read the medication leaflet?) provide insight into how well parents inform themselves before administering antibiotics. A significant percentage of parents (84%) declare that they do review instructions. This percentage is in discord with the higher percentage of people who were identified with misconceptions regarding the use of antibiotics.

The COVID-19 pandemic has had a significant negative impact on AMR, affecting both AMR surveillance activities and resistance rates worldwide, as reported by WHO. The healthcare crisis led to disruptions in monitoring programs, reduced enforcement of antimicrobial stewardship policies, and increased pressure on healthcare systems, all of which contributed to the worsening of AMR trends. Additionally, the pandemic exacerbated the inappropriate use of antibiotics, particularly in hospitalized COVID-19 patients. Studies suggest that while bacterial co-infections were relatively rare among COVID-19 patients, a large proportion still received antibiotics unnecessarily, further fueling the development of resistance [1,47,48].

### 3.3. Compliance with Antibiotic Treatment

The final set of questions (Q7, Q8, Q9, Q10, and Q11) explored parental adherence to prescribed antibiotic treatments. The results revealed several factors that contribute to non-compliance, including parental pressure on doctors to prescribe antibiotics (Q7) and seeking a second opinion when antibiotics are not prescribed (Q8). These behaviors suggest that some parents have a pre-existing expectation that antibiotics should always be given for certain illnesses, regardless of medical recommendations. The situation is sometimes more complex. While physicians often cite patient demand as a key driver of antibiotic overprescription [21,39,49], studies indicate that in pediatrics, explicit requests for antibiotics are rare [50]. Instead, a more subtle form of pressure emerges through parental behavior and questioning during consultations. When parents suggest a bacterial infection as a possible cause of their child’s illness, physicians are more likely to perceive this as an expectation for antibiotics. Similarly, when parents challenge a viral diagnosis or express skepticism about a non-antibiotic treatment recommendation, this is often interpreted as resistance and contributes to the physician’s impression that antibiotics are expected. These implicit cues can influence prescribing decisions, even in the absence of a direct request [51,52].

Another key issue was treatment adherence (Q11). Some parents admitted to discontinuing antibiotics once their child started feeling better, rather than completing the full course of treatment as prescribed. Although there is growing evidence that shorter courses of antibiotics can be just as effective as longer ones for certain infections, ongoing research aims to determine the minimum duration required to fully eradicate bacteria. However, feeling better or experiencing symptom improvement does not necessarily mean that the infection has been completely eliminated. Prematurely stopping an antibiotic course may contribute to the development of antibiotic resistance [53]. In Romania, a national survey revealed that one in four respondents believes that antibiotic treatment should be stopped once they start feeling better, rather than completing the full course prescribed by the doctor. This percentage is one of the highest in Europe. In contrast, countries such as the United Kingdom and the Nordic nations report over 90% compliance with completing antibiotic treatment as prescribed [13]. Additionally, some parents expressed hesitancy in administering antibiotics even when prescribed by a doctor (Q9) or believed that only certain antibiotics work for their child (Q10). This selective approach to antibiotic use may stem from previous experiences or misinformation, and all of these practices show a distrust in medical recommendations.

Increased levels of mistrust in healthcare providers are associated with greater reluctance to accept medical diagnoses, reduced interaction with healthcare providers, lower adherence to treatment plans, decreased willingness to adopt preventive measures, and overall poorer physical and mental health outcomes [54]. These findings emphasize the importance of improving doctor–parent communication and ensuring that parents understand why completing an antibiotic regimen is essential for treatment success. The Special Eurobarometer 478 on Antibiotic Resistance, a survey requested by the European Commission, revealed that in Romania, 40% of respondents had been prescribed antibiotics based on such a test. This figure is close to the European average [13]. A more accurate diagnosis, confirmed through the use of rapid tests (such as antigen or PCR tests), can provide significant benefits both in disease management and in reinforcing medical decisions, particularly for patients with low levels of trust in healthcare providers. By offering objective, evidence-based confirmation, these diagnostic tools can reduce uncertainty, minimize unnecessary antibiotic prescriptions, and improve patient adherence to treatment recommendations [55,56].

### 3.4. Implications and Future Directions

Addressing the issue of inappropriate antibiotic use in children requires a multifaceted approach involving healthcare providers, policymakers, educators, and parents. Strengthening regulatory measures to control the sale of antibiotics, increasing access to accurate information, and fostering a culture of antibiotic stewardship will contribute to a more responsible approach to pediatric healthcare. Ensuring that antibiotics remain an effective tool in the treatment of bacterial infections is imperative for future generations, and this can only be achieved through collective efforts to combat antibiotic misuse and resistance.

Raising parental awareness through education and public health initiatives is crucial to curbing antibiotic misuse. By enhancing the understanding of the differences between bacterial and viral infections, proper treatment protocols, and the long-term risks of antibiotic resistance, healthcare professionals can empower parents to make informed decisions. Campaigns should focus on dispelling common myths, reinforcing the importance of adhering to prescribed antibiotic regimens, and emphasizing the critical role of medical consultation before administering antibiotics to children. To effectively combat inappropriate antibiotic use in children, it is essential to recognize that healthcare providers alone may not reach all segments of the population, particularly those with lower educational levels. Studies indicate that critical health information does not always effectively reach individuals with limited health literacy, making them more susceptible to misconceptions and self-medication practices. Therefore, there is a pressing need for targeted informational campaigns tailored to these populations, using clear, accessible language, and culturally appropriate communication strategies. While healthcare providers remain central in guiding appropriate antibiotic use, their influence may be limited by time constraints, access disparities, and patient mistrust. Complementary interventions, such as community-based educational campaigns, pharmacist-led counseling, integration of antibiotic education in school curricula, and leveraging mass media, may be necessary to reach broader and more diverse segments of the population. Simplified educational materials, community-based outreach programs, and the use of visual aids or digital platforms could significantly improve awareness and understanding. Ensuring that every parent, regardless of education level, has access to reliable, easy-to-understand guidance on antibiotic use is crucial in fostering responsible healthcare behaviors and reducing the misuse of antibiotics.

A notable limitation of this study, which warrants further investigation, is that, similar to many studies in the literature, research on knowledge and perceptions regarding antibiotic use and AMR has primarily focused on patients within specific contexts or countries. Many of these studies assess only basic awareness of antibiotic indications and efficacy, offering limited insight into deeper misconceptions or behavioral patterns. There are fewer studies that explore the perceptions and practices of healthcare professionals. Addressing these gaps could provide a more comprehensive perspective on how AMR-related knowledge influences prescribing behavior and patient interactions in real-world practice.

Another important limitation of this study is the use of a convenience sampling method, which may restrict the generalizability of the findings. Since participants were selected based on availability during hospitalization rather than through random sampling, the sample may not be fully representative of the broader parent population in Romania. This approach could introduce selection bias, as individuals who agree to participate may differ systematically in terms of demographics, socioeconomic status, or health literacy from those who do not. Finally, an additional limitation concerns the survey instrument itself. While it was carefully constructed based on empirical observations and the existing literature, and reviewed by pediatricians and infectious disease specialists for content relevance and clarity, it was not subjected to formal pretesting or external validation. Although the tool was approved by the institutional ethics committee, the absence of a standardized validation process may limit the reproducibility and comparability of the results in other populations or settings.

## 4. Materials and Methods

### 4.1. Study Design and Participants

This study was designed as a quantitative, cross-sectional survey. We developed a 15 question questionnaire, structured to evaluate parental knowledge, attitudes, and practices related to antibiotic use in children. The study was conducted in the Pediatric Departments of the County Emergency Clinical Hospital Bihor over a six month period. A total of 400 participants were recruited from among primary caregivers of hospitalized children aged 0 to 18 years, admitted for both acute and chronic conditions. All participants provided informed consent prior to enrollment in the study. Participants were selected using a convenience sampling approach, and data were collected through specially designed questionnaires distributed during the child’s hospital stay. Although convenience sampling allowed for practical participant recruitment in a clinical setting, it limits the generalizability of the results to the wider Romanian caregiver population and introduces potential selection bias. The study aimed to achieve a 95% confidence level with a margin of error of ±5%, ensuring statistical reliability in assessing parental perspectives on antibiotic use. To minimize the risk of socially desirable responses, participants were assured of the anonymity and confidentiality of their answers. The questionnaire was self-administered and completed without the presence of healthcare personnel or researchers, reducing pressure to respond in a particular way. Incomplete questionnaires were excluded from the analysis. However, participants who declined to provide demographic information were included, provided that the questionnaire section was fully completed. Out of 420 distributed questionnaires, 400 were returned completed, resulting in a high response rate of 95.2%.

### 4.2. Survey Structure

The questionnaire consisted of 15 multiple-choice questions, in which participants responded with “Yes”, “No”, or “Don’t know”, grouped into three main categories: 1. Knowledge and Misconceptions—evaluating parental misunderstandings regarding the indications and effectiveness of antibiotics (Q1, Q2, Q4, Q5, Q14, and Q15); 2. Self-medication—assessing the frequency and reasons for administering antibiotics without medical consultation (Q3, Q6, Q12, and Q13); 3. Compliance—investigating adherence to prescribed antibiotic treatments, including premature discontinuation and dosage adjustments (Q7, Q8, Q9, Q10, and Q11). Additionally, information on each participant’s age, place of residence (rural/urban), and level of education was also collected.

The survey instrument was specifically developed for the purpose of this study, drawing on empirical observations as well as a review of the relevant literature on antibiotic use and common misconceptions. While the questionnaire was not previously validated or formally pretested, which constitutes a methodological limitation, its content was internally reviewed by pediatricians and infectious disease specialists to ensure clinical relevance and clarity. Additionally, the study protocol, including the questionnaire, received approval from the institutional ethics committee, further supporting the appropriateness of the instrument within the research framework.

### 4.3. Survey Questions

Have you ever heard of the term “antibiotic resistance”?Should antibiotics be administered for viral infections?Have you ever given your child antibiotics for a common cold to prevent symptoms from worsening?Do sore throat or productive coughs always require antibiotic treatment?Does the presence of fever always require antibiotic treatment?Have you ever given your child antibiotics without a doctor’s recommendation?Have you ever asked a doctor to prescribe antibiotics for your child?If a doctor does not prescribe antibiotics when you think they are necessary, do you seek a second opinion?Do you hesitate to give your child antibiotics even when a doctor explains that they are necessary?Do you believe that only certain antibiotics work for your child, or have you refused to administer an antibiotic believing it was not suitable?Have you ever stopped an antibiotic treatment early once your child started feeling better?Have you ever been able to purchase antibiotics from a pharmacy without a prescription?Do you usually read the medication leaflet before administering antibiotics?Can any type of tablet be crushed for easier administration?Can leftover antibiotic suspension be stored in the refrigerator for future use?

### 4.4. Data Analysis

Survey responses were analyzed using IBM SPSS Statistics version 26 (IBM Corp., Armonk, NY, USA) for statistical analysis, and Microsoft Excel version Microsoft^®^ Excel^®^ 2019 MSO (Version 2504 Build 16.0.18730.20122) 64-bit (Microsoft Corp., Redmond, WA, USA) for data organization and additional calculations. Descriptive statistics were used to summarize parental knowledge, attitudes, and practices regarding antibiotic use. Associations between key variables, such as education level and antibiotic misuse, were evaluated using chi-square tests and logistic regression analysis, with a significance level set at *p* < 0.05.

### 4.5. Ethical Considerations

All participants provided informed consent before completing the questionnaire. The study protocol was reviewed and approved by the Ethics Committee of the County Emergency Clinical Hospital Bihor (nr.3972/ 6 February 2025), ensuring compliance with ethical research standards. The study ensured participant anonymity, voluntary participation, and the right to withdraw without penalty. All data were stored securely and used solely for research purposes. The privacy and confidentiality of participants were strictly maintained throughout the study.

## 5. Conclusions

This study highlights the persistent prevalence of misconceptions surrounding antibiotic use among Romanian caregivers. While individual questions reveal that approximately one-quarter of respondents provide incorrect answers, a more concerning picture emerges when these misconceptions are considered cumulatively: 86% of participants hold at least one inaccurate belief regarding antibiotic administration. Notably, 42.5% of respondents reported never having heard of the concept of antimicrobial resistance, underscoring a significant gap in basic awareness.

These findings align with previous research, indicating that misconceptions are more common among individuals from rural areas and those with lower levels of education. Despite these knowledge gaps, the majority of respondents report rarely self-administering antibiotics without medical advice. However, many admit to attempting to influence physicians’ decisions regarding antibiotic prescriptions.

This pattern suggests that the prescribing behavior of healthcare professionals may play a substantial role in the broader issue of antibiotic overuse. Consequently, future research should extend beyond patient perceptions and examine the determinants influencing healthcare providers’ prescribing decisions, including perceived patient expectations, diagnostic uncertainty, and time constraints in clinical practice. At the same time, interventions aimed at improving antibiotic literacy should not be limited to clinical environments but should also incorporate community-level approaches such as education programs in schools, pharmacies, and local health campaigns. Further research should evaluate the effectiveness of these strategies, identify culturally adapted communication methods, and explore how provider–patient dynamics influence expectations and prescribing practices across different demographic and regional contexts.

## Figures and Tables

**Figure 1 antibiotics-14-00479-f001:**
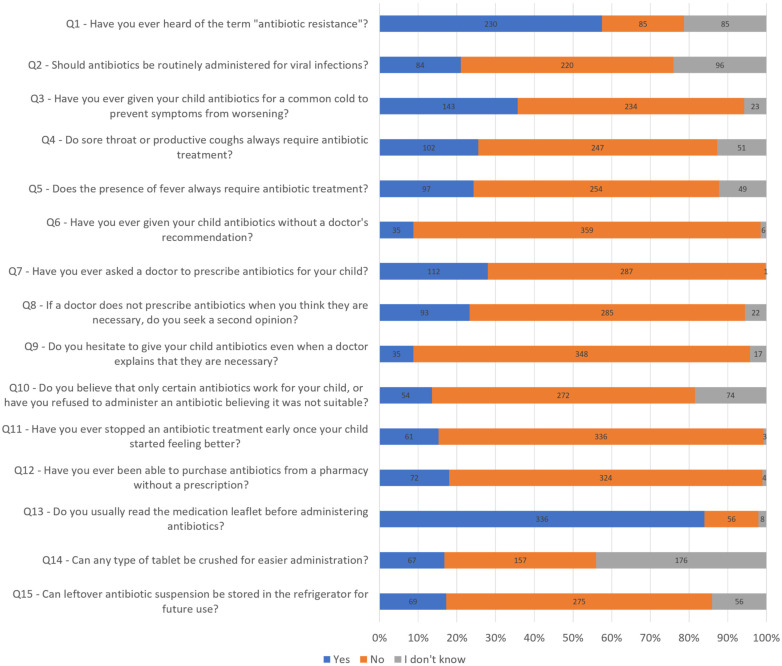
Questionnaire and caregiver responses.

## Data Availability

Dataset is available at https://zenodo.org/records/15365595 (accessed on 1 February 2025).

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
