# Peer review of "Misconceptions and Behavioral Risks in Parental Antibiotic Use on Romanian Children: A Cross-Sectional Study on Knowledge, Attitudes, and Practices"

_antibiotics, 2025, doi:10.3390/antibiotics14050479_

Round 1
Reviewer 1 Report
Comments and Suggestions for Authors
The manuscript "Parental Perceptions and Knowledge on Antibiotic Use in Romanian Children: Balancing Awareness and Risks" presents the results of a cross-sectional study on Romanian parents’ knowledge, perceptions, and behaviors regarding antibiotic use in children. Nowadays, it is a highly relevant topic in the global antimicrobial resistance (AMR) crisis. The authors offer timely insights into how parental misconceptions and educational background correlate with inappropriate antibiotic use, particularly in a high-burden country like Romania. The paper is well-structured and methodologically sound at the basic level, with actionable public health implications.
However, the article would benefit from moderate revisions to improve statistical rigor, language clarity, and methodological transparency.
The key suggestions for improvement and points requiring the authors' response are outlined below:
- Introduction
- Some region-specific data and studies should be moved earlier from the Discussion to support the rationale.
- Lines 153–169, Vague phrasing such as “clinical experience shows...” - Replace with evidence-based statements or cite a specific source to support the claim.
- Methodology
-
lines 405–416, Convenience sampling, Lack of detail about sampling method - Clarify that convenience sampling was used and explicitly address its limitations. Authors should more explicitly address this limitation in the Discussion (lines 393–403)
-
-
-
Survey instrument, lacks information on validation or pretesting (lines 418–427). If it was developed de novo, this should be acknowledged and discussed as a limitation.
-
- Statistical Analysis
- The paper relies solely on descriptive statistics and chi-square tests (Section 4.4). No multivariable analysis (e.g., logistic regression) is conducted to control for confounding between education, residence, and outcomes. Authors should either include such analysis or justify its omission.
- Discussion
- Lines 253–271 and 313–331, Redundancy and repetition of earlier findings - Condense these sections and use the space for deeper interpretation or contextualization.
- Conclusion
- Some broad generalizations (e.g., about physician behavior) go beyond what is directly supported by the data.
The English is generally understandable but would benefit from professional language editing to enhance clarity, precision, and a more scientific tone.
Author Response
Dear Reviewer,
We would like to sincerely thank you for your thoughtful and constructive feedback on our manuscript. Your comments were insightful and have significantly improved the clarity, rigor, and overall quality of our article. Below, we provide a point-by-point response to your suggestions and outline the revisions made accordingly.
- Comment: “Some region-specific data and studies should be moved earlier from the Discussion to support the rationale.”
We agree with this observation. A new paragraph (lines 65–75) was added to the Introduction to incorporate recent region-specific data, including findings from the Special Eurobarometer 478 and other national surveys, in order to better contextualize the study rationale.
- Comment: “Lines 153–169, vague phrasing such as ‘clinical experience shows...’”
This phrasing has been revised to reflect evidence-based statements, and we have cited two recent peer-reviewed sources to support these claims.
- Comment: “Lines 405–416: Clarify convenience sampling and explicitly address limitations.”
We have clarified the use of a convenience sampling strategy in the Methodology section and discussed its implications more explicitly in the Limitations subsection.
- Comment: “Survey instrument lacks validation/pretesting details.”
We have added a new paragraph in the “Survey Structure” section noting that the questionnaire was developed de novo for this study. Although it was reviewed by pediatric and infectious disease specialists for clarity and content relevance, it was not previously validated or pretested. This has also been included as a limitation.
- Comment: “The paper relies solely on descriptive statistics and chi-square tests. No multivariable analysis (e.g., logistic regression) is conducted.”
We appreciate this important suggestion. In response, we conducted a binary logistic regression analysis to evaluate the independent effects of education level and environment of origin on the likelihood of holding at least one misconception. The analysis showed that education remained a statistically significant predictor, while environment did not retain significance in the adjusted model. These findings have been integrated into the revised Results section and the Abstract.
- Comment: “Lines 253–271 and 313–331: Redundancy and repetition.”
We have reviewed and condensed these sections to eliminate unnecessary repetition and instead focused on strengthening the interpretation and contextualization of the results.
- Comment: “Some broad generalizations (e.g., about physician behavior) go beyond what is supported by data.”
We agree and have reformulated these statements to avoid overgeneralization. The revised conclusion highlights the need for further study on prescriber behavior rather than drawing firm conclusions beyond our dataset.
Once again, we thank you for your valuable insights, which have helped improve the scientific and editorial quality of our manuscript.
Reviewer 2 Report
Comments and Suggestions for Authors
The manuscript by Iuhas et al., titled “Parental Perceptions and Knowledge on Antibiotic Use in Romanian Children: Balancing Awareness and Risks”, addresses an important issue by highlighting the misconceptions parents have about antibiotic use in children. This topic is highly relevant, particularly in the context of efforts to reduce antimicrobial resistance through targeted public health education. However, there are several points that require clarification and improvement before the manuscript can be considered for publication.
The current title lacks precision and does not clearly reflect the study’s emphasis on parental misconceptions and behavioral patterns that contribute to inappropriate antibiotic use. A more suitable and descriptive title could be: “Misconceptions and Behavioral Risks in Parental Antibiotic Use for Romanian Children: A Cross-Sectional Study on Knowledge, Attitudes, and Practices.” This revised title would better convey the scope and methodology of the study.
The abstract requires clearer phrasing in some areas. For instance, in line 20, the statement that “86% of participants held at least one misconception about antibiotics” should indicate whether this figure refers to the entire sample or a subset. Additionally, lines 23–24 mention that 89.8% did not administer antibiotics without a physician’s advice, while 28% had requested prescriptions. It is unclear whether these figures represent overlapping or distinct participant groups. Clarifying this would enhance the coherence of the abstract.
In Introduction, lines 57–58, the sentence stating that “Romania is in a difficult situation, being at the intersection of two concerning trends in AMC and AMR” is ambiguous. The phrase “difficult situation” is subjective and should be replaced with a more precise description. The authors should clearly state whether Romania is characterized by high rates of antimicrobial consumption (AMC) and antimicrobial resistance (AMR), and explain how these trends affect the country and potentially the wider European region.
The Methodology section would benefit from additional detail and clarification. Specifically:
There is no mention of how the questionnaire was pretested for reliability and validity.
The rationale for choosing a convenience sample of hospitalized caregivers should be explained. The absence of community samples may affect the generalizability of the findings.
Steps taken to reduce socially desirable responses from participants should be described.
It should be stated clearly what the logistic regression analysis was used for, including the dependent and independent variables.
The inclusion and exclusion criteria for the questionnaires, as well as the response rate, should be explicitly mentioned, especially since the data collection was conducted in person.
For the result, the authors should clarify whether all questions in the survey were answered by participants, or if there were missing data. In line 116, the use of the term “substantial” is vague; it would be more informative to state whether the proportions reported represent the majority or a minority of the sample.
Discussion section, in lines 242–244, a reference should be provided to support claims about public perceptions in Romania. Additionally, the discussion would be strengthened by exploring the policy implications of the findings. For example, if healthcare providers alone are insufficient to reach all populations, what alternative or complementary strategies should be considered?
The limitations section should also be expanded. The fact that only parents of hospitalized children were included introduces a potential selection bias, which may limit the applicability of the findings to the broader population.
The conclusion should emphasize the practical implications of the findings, particularly in guiding future research and informing public health policy. The recommendation for further research should be supported by specific examples of what such research should explore, such as community-based assessments or intervention trials.
Author Response
Dear Reviewer,
We thank you for the thoughtful and constructive feedback provided on our manuscript. We appreciate the opportunity to address each of the comments and have revised the manuscript accordingly. Below are our detailed responses:
- Title Clarity
Reviewer Comment: The current title lacks precision...
We have revised the title to better reflect the scope and methodology of the study, now reading: “Misconceptions and Behavioral Risks in Parental Antibiotic Use for Romanian Children: A Cross-Sectional Study on Knowledge, Attitudes, and Practices.”
- Abstract Clarity
Reviewer Comment: The abstract requires clearer phrasing...
We have revised the Abstract to explicitly state that the 86% figure refers to the entire sample. We also clarified that the proportions regarding prescription requests and unsupervised administration refer to overlapping groups. The phrasing has been improved for clarity and coherence.
- Introduction Wording
Reviewer Comment: The phrase “difficult situation” is ambiguous...
The paragraph has been revised to state that Romania reports high rates of both antimicrobial consumption and antimicrobial resistance, and we have described the implications for both national and regional public health.
- Questionnaire Validation
Reviewer Comment: No mention of how the questionnaire was pretested...
A new paragraph has been added in the “Survey Structure” section. We clarify that the instrument was developed de novo, reviewed internally for relevance and clarity, but not externally validated or pretested—this is now also discussed in the Limitations section.
- Sampling Methodology
Reviewer Comment: The rationale for convenience sampling should be explained...
We clarified in the Methodology that convenience sampling was used for feasibility reasons and acknowledge the limitation regarding generalizability in the revised Limitations section.
- Social Desirability Bias
Reviewer Comment: Steps to reduce socially desirable responses should be described...
We have added that participants were assured of the anonymity and confidentiality of their responses to reduce the influence of social desirability bias.
- Logistic Regression Clarification
Reviewer Comment: It should be stated clearly what the logistic regression was used for...
We have now explicitly stated that binary logistic regression was used to assess the independent effects of education and environment on the likelihood of holding at least one misconception. Results are reported in the Results section and reflected in the Abstract.
- Inclusion/Exclusion Criteria & Response Rate
Reviewer Comment: These criteria should be explicitly mentioned...
We have clarified the inclusion criteria, stating that only fully completed questionnaires were included in the analysis. Demographic data were optional, but their absence did not exclude participants. This is now clearly reflected in the Methods section.
- Missing Data / “Substantial” Wording
Reviewer Comment: Clarify if all survey questions were answered...
We confirmed in the manuscript that only completed questionnaires were included. We also replaced “substantial” with a more precise description indicating the percentage and contextual relevance.
- Discussion – Reference and Policy Implications
Reviewer Comment: Add references and discuss policy relevance...
A supporting reference has been added for the public perception statement. Additionally, we expanded the Discussion to highlight the need for community-based strategies such as health education in schools and pharmacies, and to suggest policy interventions.
- Expanded Limitations
Reviewer Comment: The limitations section should also include potential selection bias...
The Limitations section has been expanded to discuss the implications of selecting only hospitalized caregivers and the potential for selection bias. We also addressed other methodological constraints including sampling and instrument validation.
- Conclusion – Practical Implications
Reviewer Comment: The conclusion should emphasize practical implications...
The Conclusion has been revised to include specific recommendations for future research, including community-based assessments, culturally sensitive interventions, and provider-focused behavioral studies.
We sincerely thank the reviewer once again for the insightful suggestions, which have greatly improved the clarity, structure, and impact of our manuscript.
Reviewer 3 Report
Comments and Suggestions for Authors
The following comments need to be addressed by the authors:
-
The resolution of Figure 1 should be improved for better clarity.
-
It would be more appropriate for the authors to list the names of antibiotics that are frequently used in these regions, along with their prevalence specific to the region or state.
-
Are the questions presented in Figure 1 aligned with WHO guidelines? The authors should clarify which guidelines were followed to develop these questions.
-
As stated in Question 2, antibiotics are not typically needed for viral infections. This point should be reconsidered or clarified.
-
The authors should be more specific about the symptoms, diseases, or infections for which these antibiotics were prescribed by doctors.
Author Response
Dear Reviewer,
We thank you for the constructive comments and the opportunity to improve the manuscript. Please find below our detailed responses to each point:
Reviewer Comment: The resolution of Figure 1 should be improved for better clarity.
- We appreciate the suggestion and have updated Figure 1 with a higher-resolution version to ensure better clarity and readability. The revised figure is now included in the manuscript file.
Reviewer Comment: It would be more appropriate for the authors to list the names of antibiotics that are frequently used in these regions, along with their prevalence specific to the region or state.
- We acknowledge the importance of understanding regional antibiotic usage patterns. However, the scope of our study was focused on parental perceptions and behaviors related to antibiotic use, rather than on clinical or pharmaceutical data. We did not collect information on specific antibiotic types or their frequency of use. Addressing this question would require access to prescription or pharmacy records, which was beyond the scope of our survey-based study. We have clarified this in the revised manuscript.
Reviewer Comment: Are the questions presented in Figure 1 aligned with WHO guidelines? The authors should clarify which guidelines were followed to develop these questions.
- The questionnaire was developed based on a review of empirical literature and known public misconceptions surrounding antibiotic use, particularly in Eastern Europe. Although there is no official WHO questionnaire, our content aligns conceptually with WHO recommendations for public education on antimicrobial resistance (e.g., WHO’s “Antibiotic Awareness” campaign). We have clarified this in the “Survey Instrument” subsection.
Reviewer Comment: As stated in Question 2, antibiotics are not typically needed for viral infections. This point should be reconsidered or clarified.
- We agree that wording should avoid ambiguity. While antibiotics are not effective against viral infections, such as the common cold or flu, this distinction may be misunderstood by some readers. We have revised the item wording for clarity to: “Should antibiotics be administered for viral infections?” We assure that the wording in Romanian was clear and the participant understood the meaning of this phrase.
Reviewer Comment: The authors should be more specific about the symptoms, diseases, or infections for which these antibiotics were prescribed by doctors.
- This study did not aim to collect clinical data or investigate specific indications for antibiotic prescriptions. Instead, our focus was on caregivers’ knowledge, attitudes, and behaviors regarding antibiotics. Gathering detailed diagnostic information would require a different study design, including medical record review or physician interviews, which was beyond the scope of this work. We have clarified this limitation in the revised manuscript.
Please let us know if further clarification is required on any of these points. Thank you again for your insightful feedback.
Round 2
Reviewer 2 Report
Comments and Suggestions for Authors
The manuscript has been improved significantly and my concerns were addressed
Reviewer 3 Report
Comments and Suggestions for Authors
The revised manuscript is acceptable